# Physical, mental and social status after COVID-19 recovery in Nepal: A mixed method study

Sashi Silwal *, Kristina Parajuli, Astha Acharya, Ajnish Ghimire, Savita Pandey, Ashok Pandey , Anil Poudyal, Bihungum Bista, Pradip Gyanwali, Meghnath Dhimal *

Nepal Health Research Council, Ramshahpath, Kathmandu, Nepal

* silwalsashi326@gmail.com (SS); meghdhimal2@gmail.com (MD)

## Abstract

### Background

Nepal has been devastated by an unprecedented COVID-19 outbreak, affecting people emotionally, physically, and socially, resulting in significant morbidity and mortality. Approximately 10% of COVID-19 affected people have symptoms that last more than 3–4 weeks and experience numerous symptoms causing an impact on everyday functioning, social, and cognitive function. Thus, it is vital to know about the recovered patient's health status and undertake rigorous examinations to detect and treat infections. Hence, this study aims to assess the health status of COVID-19 post-recovery patients in Nepal.

### Method

A descriptive cross-sectional mixed-method study was conducted in all seven provinces of Nepal. A total of 552 interviews were conducted for the quantitative study, and 25 in-depth interviews were conducted for the qualitative study among above 18 years COVID-19-recovered patients. The data was gathered over the phone through the purposive sampling method The results of a descriptive and thematic analysis were interpreted.

### Finding

The majority (more than 80%) of the recovered patients could routinely perform household duties, activities outside the home, and financial job accounting. However, a few of them required assistance in carrying out all of those tasks. Prior and then after COVID-19 infection, smoking habits reduced by about one-tenth and alcohol intake decreased by a twelve percent. A qualitative finding revealed that the majority of COVID-19 symptomatic patients experienced a variety of physical symptoms such as fever, headache, body pain, fatigue, tiredness, sore throat, cough, loss of taste, loss of smell, sneezing, loss of appetite, and difficulty breathing, while others felt completely fine after being recovered. Furthermore, there was no variation in the daily functional activities of the majority of the recovered patients, while a few were found conducting fewer activities than usual because they were concerned about their health. For social health, quantitative data indicated that more than half of the

**Data Availability Statement:** This study was conducted by the Nepal Health Research Council by the request of Nepal's Ministry of Health and Population. Data is owned by the Nepal Health

Research Council. The datasets used and/or analyzed may be made available by reasonable request via letter or email. Data access requests can be sent to the Nepal Health Research Council via email (nhrc@nhrc.gov.np).

**Funding:** This study was conducted by Nepal Health Research Council which is itself the Government Organization So the funding for the work was Government of Nepal. The funders had no role in study design, data collection and analysis, decision to publish, or preparation of the manuscript.

**Competing interests:** The authors have declared that no competing interests exist.

participants' social health was severely impacted. According to the IDI, the majority of the interviewees perceived society's ignorance and misbehavior. Family members were the most often solicited sources of support. Some participants got care and assistance, but the majority did not get affection or love from their relatives. Moreover, regarding mental health, 15 percent of participants had repeated disturbing and unwanted thoughts about COVID-19 after being recovered, 16 percent tried to avoid information on COVID-19 and 7 .7 percent of people had unfavorable ideas or sentiments about themselves. More than 16 percent of participants reported feeling some level of stress related to the workplace and home. While in-depth interviews participants revealed that COVID-infected patients who were asymptomatic didn't experience any emotional change in them but recovered patients who are symptomatic symptoms had anxiety and still being conscious of COVID-19 in fear of getting infected again Additionally, it was discovered that participants' mental health is influenced by ignorance of society, as well as by fake news posted to social media.

## Conclusion

COVID-19 infection has had an impact on physical, mental, and social well-being. Hence, to aid in the early recovery of COVID-19 patients, provision of evaluating and reporting the clinical features, early detection and management of long COVID case is needed from the local and provincial and central government of Nepal.

## Introduction

COVID-19, a global pandemic has affected many countries now and again with emerging variants along with affecting physical, mental and social wellbeing [1]. According to world-ometers.info (as of January 14, 2023)), 670,698,968 people have been infected around the world. However, only 641,835,771 have been recovered to date [2]. Nepal has reported a total of 1001046 confirmed cases of COVID-19 and 12020 deaths as of January 13, 2023. The country has seen a steady increase in cases since the first cases were reported in January 2020, with a significant spike in cases in April 2020 and a second wave in December 2020 [3]. The overall case fatality rate is around 1.2%. The government of Nepal has implemented various measures to control the spread of the virus, including lockdowns and quarantine measures, but the country's healthcare system has been overwhelmed by the number of cases. The government has set up field hospitals and other facilities to help manage the influx of patients [4].

The COVID-19 virus can cause a variety of new or ongoing symptoms in people with persistent COVID, which can last weeks or months after infection. These symptoms can get worse with physical or mental activity, according to the Centers for Disease Control and Prevention (CDC). COVID-19 has caused prolonged sickness even in people with milder symptoms, according to a recent study published in the CDC's Morbidity and Mortality Weekly Report after several weeks of recovery, more than 35% of patients experienced ongoing symptoms such as cough, exhaustion, or shortness of breath. People in Nepal are exposed to multiple sources of apparent stressors, such as the constant stream of news on COVID-19, reports of surging coronavirus cases around the world, distance from the social support system, restrictions in social life, additional concerns for basic needs supplies, and so on [5].

The death rate in Nepal is relatively low compared to other countries, but the morbidity rate is high. The country has a fragile healthcare system, with a shortage of medical supplies, staff and beds, and limited testing capacity. The government has been criticized for its slow

response to the crisis and lack of preparedness. The government also has implemented vaccination drive to reduce the spread of the virus, however, the progress of vaccination is slow as of now. Overall, Nepal is experiencing a significant impact from the COVID-19 pandemic, with a high number of cases and deaths, a fragile healthcare system, and limited resources to manage the crisis [6].

These factors, in addition to the stress caused by the COVID-19 infection, can increase patients' emotional and psychological distress during and after recovery [5]. The majority of COVID-19 recovered patients experience stress for several weeks, which normally passes quickly, although psychophysical symptoms such as fear, tension, depression, and worry may last longer. After recovering from COVID-19, individuals in Nepal may experience physical, mental, and social effects. Physically, some individuals may experience lingering symptoms such as fatigue, shortness of breath, and loss of taste or smell. They may also have long-term health complications such as lung damage or heart problems. Some COVID-19 survivors may also have to deal with chronic health issues as a result of the disease, such as increased risk of stroke or heart attack.

Because of the risk of infection and social distancing, patients who have recovered from the virus are not accepted into society [7].

Mentally, COVID-19 survivors may experience a range of emotions such as anxiety, depression, and post-traumatic stress disorder (PTSD). They may also have a difficult time adjusting to a "new normal" and may struggle to cope with the emotional and psychological impact of the disease.

Socially, COVID-19 survivors may face challenges such as social isolation, stigma, and discrimination. They may also have trouble returning to work or school and may have financial difficulties.

In Nepal, the number of mental health cases has been rising, due to the pandemic. People are struggling with isolation, economic loss, and the uncertainty of future. The government and the private sectors are working to provide mental health services and counseling.

As a result, it's essential to learn about recovered patients and conduct thorough examinations to diagnose and treat psychological, physical, and other disorders [8]. The physical, mental, and social effects of COVID-19 can be long-lasting and may require ongoing support and care. It is important for individuals and communities to come together to support COVID-19 survivors, and for the government to provide appropriate resources and services to help them recover. Hence, the goal of this research is to evaluate the physical, mental, and social health of COVID-19 patients who have recovered.

## Methods

### Study design

A descriptive cross-sectional study design with quantitative and qualitative approaches (mixed method) was adopted to assess the health status of COVID-19 post-recovery patients in Nepal. This study is based on a convergent parallel design, in which the quantitative and the qualitative data are collected and analyzed concurrently. The results are then triangulated to each other and interpreted.

### Study area and study participants

The study was carried out in all seven provinces A total of 552 COVID-19 recovered patients for the quantitative study and 25 COVID-19 recovered patients for the qualitative study were taken.

## Sample size and sampling technique

The list of COVID-19 recovered individuals was obtained from the Epidemiology and Disease Control Division (EDCD), from February 2020 to June 2020. We used the purposive sampling method in this study as it employs both qualitative and quantitative methods. From the list available from EDCD participants were selected purposively. Among the selected COVID-19 post-recovery individuals within a time frame, eligible participants aged 18 years and above, able to comprehend, receiving the call, and interested in answering the relevant questions were included in this study and hence the sample size was determined. A total of 552 COVID-19 post-recovery participants were included in the study.

For the qualitative component of the study, from the available list, In-depth Interviews (IDIs) were carried out until the information was saturated and no new information was generated with repetition, total 25 IDIs were conducted for the study, from the available list, sampling was done based in the theory of saturation.

## Study variables

In our study dependent variable were COVID-19 recovered patient and the independent variable included gender (male, female), age(18 to 60 years above), marital status (married, divorced/separated, single), Ethnicity (Hill Brahmin, Hill Chhetri, Terai Bhramin, Hill Dalit, terai Dalit, Newar, other Hill Janjati, Terai Janjati, Other Terai caste and others) Religion (Hindu, Buddhist, Muslim, Christian, Kirat), Education, (illiterate, literate but no formal, education (primary, secondary, higher) Occupation (healthcare worker, non-government job, housewife, government job, abroad employment, army/ police, student, labor work, self-business, others) travel history (Abroad, inter province, inter municipality, not traveled), place of stay (home isolation, hospital, Quarantine), health status, (physical health, social health, mental health).

## Data collection tools

For the survey of COVID-19 recovered patients, several literature review was done and based on the information required and defining the target population, a structured questionnaire as well as IDI guidelines was designed which was classified into 4 categories (Socio demographic, physical wellbeing, mental wellbeing and social wellbeing). Both the qualitative and quantitative tools were designed in a way that could cover key issues to elicit experience of COVID-19 recovered patients in terms of physical, mental and social well-being and telephonic interview was done to generate the information.

## Data collection techniques

A telephonic interview for both quantitative and qualitative was taken by 21 enumerators. Structured questionnaires and In-depth Interview guidelines were developed in English and translated into the local language (Nepali) by the experts and questions were rephrased into a language that participants understand. A full translation of the IDI was prepared on the same day. Developed translations were discussed thoroughly with the research team to identify confusing issues that need further exploration. Issues that need clarification or exploration were covered in the following interviews.

## Pretesting

Pretesting was conducted among ten COVID-19-recovered individuals from the list available from EDCD. Validity was assured by consultation with an expert on the concerned subject.

### Inclusion criteria

Individuals recovered at home, quarantine center or hospital-admitted patient.
Individuals who are 18 years and above and willing to answer via phone call.

### Exclusion criteria

Patient unreachable by a phone call at a time of interview and data collection period.
Patient unwilling to respond.

### Data management and analysis

**Quantitative.** Data was entered into Ms-excel and it was cleaned. The clean version of dataset was than exported to Statistical Package for the Social Sciences (SPSS) version (16), and descriptive analysis which includes frequency, percentage and cross-tabulation was done.

**Qualitative.** The recorded audio file was translated into English in the same day of the interview and checked for accuracy thoroughly for consistency. The analysis was done manually and broadly following the six stages of thematic analysis. Getting acquainted with the data, (ii) generating initial codes, (iii) searching for themes and patterns, (iv) reviewing themes and patterns, (v) defining and (vi) naming themes. Thus, themes were finalized, and thematic analysis was done.

**Ethical consideration.** Ethical approval was taken from the Ethical Review Board (ERB) of the Nepal Health Research Council (NHRC) with reference number 569/2020. With the assurance of confidentiality and anonymity, the study objectives, methods, risks, and benefits, the needs of this study, and the expected outcome of this study were well explained to all the participants. Data was collected via phone interview by taking verbal consent.

## Quantitative results

### Socio-demographic characteristics

Table 1: Socio-demographic Characteristics showed that male participants were twice the female participants. Most (29%) of them were between age group of 21–30 years. The majority (70.5%) of them were married, and most (23.6%) of them were Hill Brahmins. Also, the majority (90.8%) belonged to the Hindu religion. Almost half of them (47.3%) had a higher education degree (Bachelor's and above). Similarly, most (42.7%) of them had military or police backgrounds while only (2%) of them were students.

### Travel history and place of stay

Table 2: Travel history and place of stay reveals that most (44.6%) of the recovered COVID-19 participants having inter-municipality travel history. Whereas, almost 30 percent had traveled overseas. In terms of place of stay for recovered participants, the majority (32.6 percent) stayed at home in isolation, while 22.3 percent stayed hospital. Similarly, 18.1 percent of them remained in quarantine while infected.

### Co morbid condition of COVID-19 recovered participants

Table 3: Co-existing disease of COVID-19 recovered participants that among the total recovered participants in the study, 70 of overall recovered research participants had co-morbid conditions. Out of which, more than half (54.3%) of them had diabetes, 31.4 percent had heart disease, and 7.1 percent had asthma.

**Table 1. Socio-demographic characteristics.**

| Variables | Frequency (N = 552) | Percent (%) |
|---|---|---|
| **Gender** | | |
| Male | 378 | 68.5 |
| Female | 174 | 31.5 |
| **Age** | | |
| 18–20 | 23 | 4.0 |
| 21–30 | 230 | 29.0 |
| 31 to 40 | 156 | 28.3 |
| 41 to 50 | 88 | 15.9 |
| 51 to 60 | 48 | 8.7 |
| More than 60 | 17 | 4.5 |
| **Marital Status** | | |
| Married | 389 | 70.5 |
| Divorced/Separated | 2 | 0.4 |
| Single | 6 | 1.1 |
| **Ethnicity** | | |
| Hill Brahmin | 130 | 23.6 |
| Hill Chhetri | 106 | 19.2 |
| Terai Brahmin/ Chhetri | 56 | 10.1 |
| Hill Dalit | 36 | 6.5 |
| Terai Dalit | 94 | 17.0 |
| Newar | 20 | 3.6 |
| Other hill janajati | 91 | 16.5 |
| Terai janajati | 14 | 2.5 |
| Other terai caste | 2 | 0.4 |
| Others | 3 | 0.5 |
| **Religions** | | |
| Hindu | 501 | 90.8 |
| Buddhist | 26 | 4.7 |
| Muslim | 16 | 2.9 |
| Christian | 8 | 1.4 |
| Kirat | 1 | 0.2 |
| **Education** | | |
| Illiterate | 21 | 3.8 |
| Literate but no formal schooling | 26 | 4.7 |
| Primary education | 80 | 14.5 |
| Secondary Education | 164 | 29.7 |
| Higher | 261 | 47.3 |
| **Occupation** | | |
| Healthcare Worker | 48 | 8.7 |
| Non-government job | 62 | 11.3 |
| Housewife | 26 | 4.7 |
| Government Job | 41 | 7.4 |
| Abroad Employment | 14 | 2.5 |
| Army, Police | 236 | 42.7 |
| Student | 11 | 2 |
| Labor Work | 37 | 6.7 |
| Self-Business | 62 | 11.3 |
| Others | 15 | 2.7 |

**Table 2. Travel history and place of stay.**

| Travel history | Frequency (N = 552) | Percent (%) |
|---|---|---|
| Abroad | 165 | 29.9 |
| Inter province | 68 | 12.3 |
| Inter District | 66 | 12.0 |
| Inter Municipality | 246 | 44.6 |
| Not traveled | 7 | 1.2 |
| **Place of stay during COVID-19 infection** | | |
| Home isolation | 180 | 32.6 |
| Hospital | 123 | 22.3 |
| Quarantine | 100 | 18.1 |
| Not revealed | 149 | 27.0 |

**Table 3. Co-existing disease of COVID-19 recovered participants.**

| Co-Morbidity | Frequency (N = 70) | Percent (%) |
|---|---|---|
| Diabetes | 38 | 54.3 |
| Heart Disease | 22 | 31.4 |
| Chronic Lung Disease | 2 | 2.9 |
| Chronic Liver Disease | 2 | 2.9 |
| Asthma | 5 | 7.1 |
| HIV Infection | 1 | 1.4 |

**Table 4. Co-current weight and smoking and alcohol consumption behavior.**

| Smoking and Alcohol Consumption behavior | Before COVID-19 N (552) | After COVID-19 N (%) |
|---|---|---|
| Smoking | 113(20.5) | 65(11.8) |
| Alcohol | 150(27.2) | 86(15.6) |

Table 4: Co-Current weight and smoking and alcohol consumption behavior provides information about the smoking and alcohol consumption behavior of the recovered individuals before and after COVID -19.

Among the post-recovery participants, current smokers and users of alcohol accounted for 27.2 percent and 15.6 percent respectively. However, after being infected with COVID-19, the percentage of smokers dropped to 20.5 percent, and the percentage of alcohol users dropped to 11.8 percent.

Table 5: Sociodemographic characteristics of COVID-19 recovered patients IDI participants displays that out of 25 participants, the majority were married. Most persons were between the ages of 31 and 40. Majority belonged to Janjati in terms of an ethnic group. In regards to education, more than half of the participants obtained higher secondary levels. Moreover. almost half of the participants were army/police.

## Quantitative and qualitative results

### Physical health status of COVID-19 recovered participants

Table 6: Physical health status of COVID-19 recovered participants illustrates that the majority (89.1%) of the COVID-19 individuals who had recovered had generally good health. Despite

**Table 5. Sociodemographic characteristics of COVID-19 recovered patients IDI participants.**

| Variables | Frequency (N = 25) |
|---|---|
| **Marital status** | |
| Married | 19 |
| Unmarried | 6 |
| **Age** | |
| 20–30 | 11 |
| 31–40 | 12 |
| Above 41 | 2 |
| **Caste** | |
| Janajati | 10 |
| Chhetri | 6 |
| Brahmin | 8 |
| Dalit | 1 |
| **Education** | |
| Secondary | 4 |
| Higher secondary level | 14 |
| Bachelor and above | 7 |
| **Occupation** | |
| Police/Army | 12 |
| Student | 2 |
| Goldsmith | 1 |
| Foreign employment | 2 |
| Indian army | 3 |
| Office job | 3 |
| Teacher | 1 |
| Farmer | 1 |

being recovered from COVID -16, 10.5 percent of recovered participants stated having few problems and two of them had complicated health issues.

Table 7: Cross-tabulation between physical health and physical activities of the past 1 month shows more than 80 percent of recovered COVID-19 participants performing all normal physical activities, such as household chores, other activities outside the home, and other professional activities. However, it was found that 6.2 percent of recovered individuals were reliant on completing housework. Similarly, 7.4 and 8.2 percent of the recovered individuals were found to be dependent on activities outside the house and on financial work, respectively.

During the qualitative interview, out of 25 post-COVID-19 recovered patients, twenty (20) COVID-19 recovered patients were fit and fine. They shared that they were completely fit and normal as previous and can do all regular activities.

**Table 6. Physical health status of COVID-19 recovered participants.**

| Current health condition) | Frequency(N = 552) | Percent (%) |
|---|---|---|
| Normal | 492 | 89.1 |
| Few Problems | 58 | 10.5 |
| Complicated | 2 | 0.4 |

**Table 7. Cross-tabulation between physical health and physical activities.**

| Physical well-being | Physical activities (N = 552) | | |
|---|---|---|---|
| | Doing household chores N (%) | Doing activities outside home N (%) | Finance related work N (%) |
| Normal | 478(86.6) | 483(87.5) | 466(84.4) |
| Has difficulty, but does by self | 21(3.8) | 19(3.4) | 25(4.5) |
| Requires assistance | 19(3.4) | 9(1.6) | 16(2.9) |
| Dependent | 34(6.2) | 41(7.4) | 45(8.2) |

*"Right now, I have been eating well. It's better than before. I am the same as I was before"*

- 35yrs, Farmer, who arrived from abroad, in Sudurpashchim Province.

Meanwhile, a few of them experienced a loss of appetite and tiredness, and some experienced headaches and eye pain. Furthermore, an itchy nose, sneezing frequently, suffering from cough and cold, irritation in the throat, and loss of smell and taste were common physical problems experienced even after the recovery from COVID-19.

*"I slightly felt a loss of smell and loss of taste."*

- 26yrs, health Officer, Inter-province, Lumbini province

*"I get tired easily. In the beginning, days while I discharge there was loss of appetite for 2–3 days."*

Furthermore, qualitative results revealed that some recovered individuals did fewer physical activities than normal. A few people did not take part in any of the activities because they were terrified of reinfection. Few persons claimed to have quit drinking when infected with COVID-19.

"My duty already resume but only inside the compound because of COVID -19"

-37 years living within the country, police, Bagmati province

*"I attend my computer lessons, rest throughout the day, and devote attention to my academics in the evening. But now I'm going to concentrate on my football practice.*

- 20yrs, living within the country, teacher, Sudurpaschim province.

## Social health status of COVID-19 recovered participants

Table 8: Treated place and social health status of COVID-19 recovered participants displays the social health status of COVID-19 recovered participants, where 82 COVID-19 recovered

**Table 8. Treated place and social health status of COVID-19 recovered participants.**

| Place of stay | Social health (N = 470) | |
|---|---|---|
| | Low N (%) | High N (%) |
| Home isolation | 79(47.3) | 88(52.7) |
| Hospital | 55(55) | 45(45) |
| Quarantine Centers | 41(47.7) | 45(52.3) |

participants stated that their society was unaware of their COVID-19 infection. Thus, social health status was determined among 470 individuals.

The average cutoff point was 12.4, while the minimum and maximum values were 1 and 26, respectively. Low social health status was defined as a value below 12.4; high social health status was defined as a value over 12.4. It was revealed that 51.7 percent of recovered participants had a high social health status, whereas 48.3 percent had a low social health status.

It was found that more than half (55%) of recovered individuals who stayed in the hospital had poor social health, followed by quarantine centers (47.7%) and home isolation (47.3%). However, 117 of the recovered participants did not reveal their residence.

Participants from the in-depth interview felt ignorance and misbehave from society. Whereas, few didn't experience any kind of hatred from society.

*"Here in my society, they do not know that I am corona positive. I have faced ignorance and embracing moments from society."*

- 32 years, Police, Living within the country, Madhesh province

*"In our village /society does not look corona infected people in a positive way they said that we shouldn't go near "infected person and contact them."*

- 49 years, Arrived from abroad, foreign employment Province-1.

Only a few participants stated that their society had responded positively and, were not subjected to any type of societal prejudice. While, others choose not to tell society about their infection to avoid any social indifference, believing that they would have to suffer even more if society became aware of their condition.

*"There is a lack of COVID awareness among the educated people also and they misbehave the corona positive people."*

- 27 yrs, Police, Living within the country, Bagmati Province

*"At that time my mother had typhoid but community people thought it was corona and were afraid of it but relatives and community people were afraid to come to our house."*

- 32 yrs, health office, Travel inter-province, Province-1

A common view was revealed from the majority of the participants as they didn't receive any support, affection, or care from their relatives, while a few received concerns from their near ones. They explained that they were neglected since their relatives were aware of their infection While family members were found to be key sources in providing help, family members were more worried about their closest ones. But some participants didn't prefer to inform their mothers and fathers as it could make them mentally weak.

*"Relatives used to be afraid of me after hearing I was infected, even some people used to be afraid of having phone calls, after my corona positive report relatives and community people were afraid to come in our house."*

- 32yrs, Police, Living within the country,Madhesh province.

*"My sister, brother, mother, father, maternal uncle, and aunt were more focused on the fact that I returned alive and well."*

- 20yrs, Teacher, Living within the country, Sudurpaschim province.

*"I didn't inform my mom and dad. I informed my brother about it. I didn't want to bother my mom and dad mentally as there are so many bad news and rumors in media regarding corona infection like kills people."*

- 30 years, Government officer, Living within country, Madesh province

## Mental health status of COVID-19 recovered participants

Table 9: Mental Health Status depicts that almost 30 percent were intensely worried about monetary reasons because of being jobless, followed by 21.4 percent experiencing work/business stress. Correspondingly, 19.2 percent were also stressed as they were staying at home even after recovery from COVID-19. In addition, 15 percent of participants experienced recurrent thoughts and were disturbed by undesirable thoughts. Also, a few (7.7%) participants had negative thoughts toward themselves and others.

During in-depth interviews it was found that the majority did not experience any emotional changes within themselves, but those participants who were symptomatic during the COVID-19 infection had anxiety, despair, and fear of reinfection as they had mindset that they haven't got the rid of the virus completely. Similarly, few people said that being ignorant also affected their mental health. Apart from this, a few individuals noted that unreliable sources of breaking news on COVID-19 were also a factor that increased the level of stress.

*"Sometime I doubt I get corona again when I have a mild headache or sore eyes or pain in chest so it impacts my mental health."*

- 30 years, Police, living within the country, Bagmati province.

*"I have felt the difference in my mental status. It not due to corona. But due to the behaviour of society towards the corona infected people. I have felt somehow negative."*

-30 years,Government officials,living within the country,Madesh province

*"While watching different kinds of news regarding COVID-19, I heard lungs will damage. Overall a lot of people felt the same thing and become scared."*

- 27 yrs, Police, living within the country, Bagmati Province

Some of the interviewees said that the COVID-19 pandemic had a significant impact on their socioeconomic position. They said that a decrease in their daily and monthly wages forced them to burrow loans to cover their fundamental needs. Despite the fact that some of them were adapted to the scenario since they received daily income from their separate

**Table 9. Mental health status.**

| Mental health status | (N = 552) | |
|---|---|---|
| | Yes N (%) | No N (%) |
| Repeated disturbing and unwanted thoughts about the COVID-19 outbreak | 83(15) | 469(85) |
| Trying to avoid information or reminders about the COVID-19 outbreak | 89(16.1) | 463(83.9) |
| Negative feelings/thoughts towards themselves and others | 11(7.7) | 509(92.3) |
| Work/business Stress | 118(21.4) | 434(78.6) |
| General stress at home | 106(19.2) | 446(80.8) |
| Severe financial stress/Due to unemployment | 163(29.6) | 389(70.5) |

employment, they were also concerned about others who had poor economic standing because there was no socioeconomic output and people's financial situations were getting worse.

> *"We receive less salary than before. At this time, we need more diet foods and a 2-time meal at the office isn't sufficient for us. We used to borrow money from friends to buy nutritious food."*
>
> - 37 yrs. living within the country, police, Bagmati Province

> *Infected people are treated inhumanely. People will not hire people who have been infected with COVID -19. Their economic progress will be hampered. It would be great if the government become worried about this. It is possible to increase economic output.*
>
> "- 30 yrs., living within the country, Government Officer, Madhesh Province.

## Coping mechanism

The majority of participants stated that exercising, using social media, listening to religious music, watching motivational films, sharing problems, dancing, and reading books were the most effective ways to cope with stressful situations. They also stated that self-motivation, such as the optimistic and innermost feeling of being emotionally and physically healthy, is their source of strength in coping with the panic situation.

> *"I am young, I convince myself that nothing will happen to me."*
>
> - 27 yrs, Police, living within the country, Bagmati Province

## Cross-tabulation

### Cross-tabulation between work/ business stress with educational status and place of stay

Table 10: Work/ business stress with educational status and place of stay displays that the majority of the participants from different education levels didn't feel the work/business-related stress. However, 19 percent of illiterate participants had work-related stress. On contrary, work/business stress was found highest (38.5%) among literate participants followed by

**Table 10. Work/ business stress with educational status and place of stay.**

| Educational Status | Work/Business stress (N = 552) | |
|---|---|---|
| | Yes N (%) | No N (%) |
| Illiterate | 4 (19) | 17(81) |
| Literate but no formal schooling | 10 (38.5) | 16(61.5) |
| Primary education (1–7) | 26 (32.5) | 54(67.5) |
| Secondary Education (8–12) | 31 (18.9) | 133(81.1) |
| Higher (Bachelor and Above) | 47 (18) | 214(82) |
| **Place of stay for treatment** | | |
| Home isolation | 42 (23.8) | 137(76.2) |
| Hospital | 19 (15.4) | 104(84.6) |
| Quarantine | 20 (20.2) | 79(79.8) |
| Not revealed | 37 (24.2) | 114(76.0) |

32.5 percent of participants with primary education. Also, 23.8 percent of participants staying at home in isolation had work/business stress compared to other places of stay.

## Cross-tabulation between treatment place and general stress at home

Table 11: Cross-tabulation between treatment place and general stress at home displays that about 22 percent of COVID-19 recovered participants from home isolation had experienced general stress, followed by the 21.3 percent of recovered participants who had stayed in quarantine.

## Cross-tabulation between treatment place and financial stress

Table 12: Cross-tabulation between the place of stay for treatment and financial stress illustrates that 43.4 percent of COVID-19 recovered participants who stayed in quarantine had experienced financial stress. In contrast, 18.7 percent of COVID-19 recovered participants who had stayed at the hospital had the lowest level of stress in comparison to other places of stay.

## Discussion

This study is first of its kind to evaluate the comprehensive health status of COVID -19 recovered patients. Mainly, our study findings demonstrated physical, emotional, and social well-being of COVID-19 post-recovery patients in Nepal. Overall, a majority (more than 80 percent could function daily activities and few required assistance, almost half of the participants (48.3 percent) had a low social health status. less proportion of the recovered individuals had repeated disturbing thought and negative feelings towards themselves.

After recovery from COVID-19 individuals who had symptoms during the time of infection experienced a variety of physical symptoms such as fever, headache, body pain, fatigue, tiredness, sore throat, cough, loss of taste, loss of smell, sneezing, loss of appetite, and difficulty breathing. While others were found doing fewer activities than normal. The majority of

**Table 11. Cross-tabulation between treatment place and general stress at home.**

| Place of stay for treatment | General Stress at home (N = 552) | |
|---|---|---|
| | Yes N (%) | No N (%) |
| Home isolation | 40(22.2) | 140(77.8) |
| Hospital | 15(12.2) | 108(87.8) |
| Quarantine Centers | 21(21.3) | 78(78.8) |
| Not revealed | 30 (20.1) | 120(80.1) |

**Table 12. Cross-tabulation between the place of stay for treatment and financial stress.**

| Place of stay for treatment | Financial stress (N = 552) | |
|---|---|---|
| | Yes N (%) | No N (%) |
| Home isolation | 49 (27.2) | 131 (72.8) |
| Hospital | 23(18.7) | 100 (81.3) |
| Quarantine | 43 (43.4) | 56 (56.6) |
| Not revealed | 48 (31.8) | 102 (68.5) |

interviewees saw society as ignorant and misbehaving. Hence domain wise health status of the recovered patient was briefly presented below.

Our survey examined the key factor that the majority (more than 80%) of the recovered participants were able to perform various forms of physical activities in the same manner as before and was also able to undertake activities outside the home routinely, such as agricultural work, purchasing commodities, and animal husbandry. Furthermore, 3.8 percent of the participants claimed that despite the difficulty, they handle all of their household activities on their own, while 3.4 percent needed help. In addition, 7.4 percent of the participants were reliant on doing outside work. Similarly, those working in finance, such as in a business, office, or labor force, were able to carry out their tasks only. Nevertheless, 8.2 percent of those surveyed were dependent on doing financial-related work. Our qualitative analysis showed that compared to their prior everyday functioning, several of the participants were engaged in less functional activities.

Another study conducted among the COVID 19 recovered participants in Nepal revealed that more than half of the participants (56.6%) reported having no functional limitations during the post-COVID-19 recovery condition after RT-PCR negative status, while 46 (43.4%) reported having some degree of functional limitation in their daily life [9] which is somehow consistent to our findings.

Also the study conducted in Nepal during the post-COVID phase showed aligned to our findings and stated that fatigue was the most common persistent symptom, with 34% experiencing fatigue after 60 days and 28.3% even after 90 days from the onset of symptoms [9].

Concurring to the finding of our study a lower functional level was documented in (47.5%) of individuals in a cross-sectional entitled "Functional status after 6 months following COVID-19." It was discovered that due to persistent symptoms, i few participants periodically minimize typical activities, have limitations in everyday life, are not able to perform all usual activities, and are dependent on another person [10]. Whereas a quantitative report conducted via phone among participants after discharge from ICU and hospital wards in Europe also showed that 30.9% of the ward patients had experienced worsen mobility, 17.6% had difficulty in self-care and 36.8% had worsen usual activities [11]. In line with our study, a study "Low physical functioning and impaired performance of activities of daily life in COVID-19 participants who survived hospitalization "conducted in Italy also showed that feeding, personal toileting, bathing dressing/undressing, and toilet usage, controlling bladder and bowel, mobility, were all identified among some participants who survived the hospitalization due to COVID-19 [12]. Similar to our findings, a study conducted in the United States demonstrated that participants who were discharged from the hospital after receiving COVID-19 saw t improvements in symptoms and function, as well as a decrease in activity-of-daily-living dependency [13].

In our qualitative finding of our study few participants reported occurrence of physical symptoms even after recovering from COVID-19. The majority of them reported a loss of appetite, exhaustion, headaches, and tiredness after a few workouts, tiredness, eye pain, nose, sneezing frequently, cough, cold, irritation in the throat, and loss of smell and taste.

Similar to our findings "A Mixed Methods Study of Functioning and Rehabilitation Needs Following COVID-19 conducted in Nepal symptoms after the COVID -19 never went away, fatigue, and recurrent headaches were among the symptoms that were frequently mentioned in their conversations [14].

Whereas, a study 'Persistent Symptoms in Participants After Acute COVID-19" highlighted only 18 (12.6%) people didn't have COVID 19–related symptoms, while 32 percent had one or two, and 55 percent had three or more. There were no signs or symptoms of acute illness in

any of the participants. In 44.1 percent of cases, participants' quality of life had deteriorated. Fatigue (53.1%), dyspnea (43.4%), joint pain (27.3%), and chest pain (27.1%) were found among the participants [15].

Similarly in a study, conducted in UK' Persistent symptoms after COVID-19: a qualitative study of 114 "long COVID" participants and draft quality principles for services" many participants stated having a significant decline in their capacity to accomplish fundamental daily tasks and also experienced symptoms that are variable and frequently relapsing-remitting, a poor prognosis, and elicited negative reactions from friends, family, and employers [16]. But in contrast to our study, an observational cohort study found that individuals' physical and mental health was worse in their post-COVID state (43.8, SD 9.3; mental health 47.3, SD 9.3) compared to their pre-COVID condition (54.3, SD 9.3; 54.3, SD 7.8, respectively), Almost three-quarters of participants reported continuing shortness of breath after being discharged from the hospital for more than a month [17].

## Mental health

Our study, showed 15 percent of participants experienced recurrent thoughts and were disturbed by undesirable thoughts. Besides this, 7.7 percent had negative thoughts towards themselves and others. Symptomatic participants, on the other hand, reported feelings of unease, depression, and terror of reinfection. Also, more than one fourth of participants were extremely concerned about their financial situation as a result of joblessness, followed by 21.4 percent experiencing work/business stress. Similarly, even after recovery from COVID-19, 19.2 percent of respondents remained stressed while staying at home.

Consistent to our findings a mixed method study conducted in Nepal revealed that some of the participants experienced problems with concentration and memory, lack of energy and motivation [16]

Congruous with our findings, a mixed study in India showed about two-thirds of respondents had lost household income, and one-third had lost jobs. However, they cope with the situation as the participants had received financial support from the government [18]. It was found from our findings that before the negative report, people infected with COVID-19 had to separate themselves, which affected their monthly income, as reported by a few of the participants. They also stated that having their daily monthly income reduced caused them to borrow money to cover their basic needs. Unlike this study, a similar study in Kenya did not show any evidence of households coping through increased borrowing or withdrawing savings. Instead, households lent out less money, postponed loan repayments, and deposited fewer savings. Households also significantly reduced expenditures, especially on schooling and transportation, in line with preventive measures such as school closures and travel restrictions [19].

The study "The Psychological Experience and Intervention in Post-Acute COVID-19 Inpatients" in 2021 found that among a total of 181 people admitted to hospitals, almost half of the participants were subjected to psychological evaluations. Acute stress disorders (18.6%), nervous and demoralization symptoms (26.7%), depression (10.5%), and troublesome grief (18.6%) were the most common psychological concerns (8.1%). Among them, 7% showed good adaptation, and 11.6% were supported only with video calls, whilst a CBT psychological treatment was fundamental for 75.59% of them [20] which is consistent with our findings. Correspondingly, in the next study conducted via. phone among participants after discharge from the intensive care unit and psychological discomfort (65.6 percent in the ICU and 42.6 percent on the ward) the prevalence of PTSD symptoms was shown to be twice as high in ICU patients as it was inward patients. The ICU group exhibited higher post-illness cough and vocal changes, which corresponds to a recent study describing the care demands of COVID-19

patients [11]. Similarly, a study was conducted on Post-Traumatic Stress Disorder in Patients after Severe COVID-19 Infection. It was found that 115 people were found to have PTSD (30.2%). Depressive episode (17.3%), hypomanic episode (0.7%), generalized anxiety disorder (7%), and psychotic disorder (0.2%) were among the other diagnoses in the whole sample [16]. Another study consistent with the present study was done by Delfina Janiri et al. on "Psychological Distress after COVID-19 Recovery". The findings revealed that 10% of the respondents had serious psychological distress, 4.5 percent had a functional impairment, and 7% had significant mental preoccupation [21].

## Social health

This study described the participants; perspectives on the stigma experience which revealed that almost half of participants have poor social health status. The qualitative finding also revealed most of the recovered patients had a high level of social stigmatization. The majority of the participants felt ignorance and misbehave by society and did not experience any affection and support from their relatives. Few of the COVID-19 infected participants realized that being ignored by their relatives and society was due to their fear of COVID-19, which they thought might affect them as well. They further stated that some of the relatives are frightened of receiving even phone calls. But in contrast, few recovered patients disclosed never being exposed to social discrimination. Instead, society's people behave positively. While a few others preferred not to inform society about their infection to avoid possible societal ignorance, this impact is caused by society's false impression of COVID-19 infection and a lack of understanding about it. Similar stigmatization has been found in Harare, Zimbabwe, according to the qualitative study, the corona survivor revealed the road in front of his house was named corona road and some people even preferred to avoid the road due to fear of infection [22].

In contrast to our findings the study held in Nepal entitled "Lived Experiences of COVID-19 Survivors in Nepal: A Phenomenological Analysis stated that, seven out of ten (7/10) of the participants reported responded feeling of stigmatizing responses and discrimination from health care workers during hospital stay [23].

More over qualitative research conducted in China, participants reported feelings of stigmatization, which is consistent with our findings. They reported that stigmatization began when they were initially suspected of having in contact with the infection and was continued even after they had recovered from the disease. Some disease-free family members have also been treated unfairly. Moreover, some participants chose to keep their history of COVID-19 confidential. The reason for non-disclosure was fear of societal indifference as they would have to suffer even more. Another similar study in China showed the same findings as it revealed most recovered participants did not disclose their history. The reason for it is mainly because of fear and lack of understanding among the public. Lack of knowledge and misinformation are contributing factors to stigma. Hence, Reliable information on disease prevention, treatment options, and accessibility of health care should be disseminated on social media by governments, the communities, media, and key influencers as potential methods for combating stigma [24]

Our study assessed the physical, social and mental health status of COVID 19 post-recovery patients which helped to identify the persistence of symptoms of COVID-19. However, the study may have several strengths as well as limitation. Firstly, the study included participants staying at home isolation, quarantine centers and hospitals as well so the information generated was representative. Second, the selection of participant was random and since no face-to-face interview was conducted there was limited hesitation by the participants to provide the required information. Also, it provides evidence-based recommendations to the government

of Nepal on improving the physical, mental and social health status and socio-economic productivity during COVID-19 and other such pandemic. However, it was a cross-sectional telephonic study, thus limiting the causal relationship. And, due to inaccurate contact number and an inaccessible network, the data was not representative which has been the limitation of our study.

## Conclusions

Overall, it appears that the majority of individuals who have recovered from COVID-19 were able to resume their normal daily activities, although a small percentage required some form of assistance. A decrease in tobacco and alcohol consumption was also observed among the study participants after recovery. Common physical symptoms included fever, headache, and body pain, although some individuals experienced no symptoms at all. The social health of more than half of the participants were significantly impacted, with many seeking supports from their families and society but not always receiving it. Additionally, a small percentage of individuals reported experiencing disturbing thoughts about COVID-19 after recovery and a similar proportion reported feeling stress related to their work and home environments. It is worth noting that the mental health of these individuals may have been affected by both societal ignorance and misinformation, as well as the proliferation of fake news on social media. Hence this finding will contribute to developing specific intervention policies and also could find out the ways in improving mental, social and physical health status of people for any pandemic that may happen in the future or for any other infection that could affect the physical, social and mental health of the people.

## Supporting information

**S1 Questionnaire.**
(DOCX)

## Acknowledgments

The authors would like to thank all the individuals and organizations that provided technical support for the design and implementation of the main survey.

## Author Contributions

**Writing – original draft:** Sashi Silwal, Kristina Parajuli, Astha Acharya, Ajnish Ghimire, Savita Pandey, Ashok Pandey, Anil Poudyal, Bihungum Bista.

**Writing – review & editing:** Pradip Gyanwali, Meghnath Dhimal.

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
