## [Decision Letter · Decision Letter 0]

29 Nov 2022

PONE-D-22-22288Physical, Mental and Social status after COVID -19 Recovery in Nepal: A Mixed Method StudyPLOS ONE

Dear Dr. Silwal,

Thank you for submitting your manuscript to PLOS ONE. After careful consideration, we feel that it has merit but does not fully meet PLOS ONE’s publication criteria as it currently stands. Therefore, we invite you to submit a revised version of the manuscript that addresses the points raised during the review process.

Please address the methodological issues raised by both reviewers.  These revisions are required for consideration for publication.

We look forward to receiving your revised manuscript.

Kind regards,

Rosemary Frey

Academic Editor

PLOS ONE

Journal Requirements:

"This study was conducted by Nepal Health Research Council which is itself the Government Organization  So  the  funding for the work was Government of Nepal. "

Reviewers' comments:

Reviewer's Responses to Questions

**Comments to the Author**

1. Is the manuscript technically sound, and do the data support the conclusions?

Reviewer #1: Partly

Reviewer #2: Partly

2. Has the statistical analysis been performed appropriately and rigorously? 

Reviewer #1: I Don't Know

Reviewer #2: Yes

3. Have the authors made all data underlying the findings in their manuscript fully available?

Reviewer #1: Yes

Reviewer #2: No

4. Is the manuscript presented in an intelligible fashion and written in standard English?

Reviewer #1: Yes

Reviewer #2: Yes

5. Review Comments to the Author

Reviewer #1: Dear Authors,

Thank you for raising the issues of physical, mental and social status after COVID 19 recovery in Nepal. This is really an important piece of work but requires a lot of revision and rewriting. I have a few comments on this paper.

1. Your way of referencing is varied throughout the paper. Please refer to PLOS one referencing guideline for details. Please make consistency for referencing throughout the paper.

2. Introduction section is completed within a paragraph which still can provide a lot of background information about Nepal. Though little information of Nepal is given. Still it is important to discuss how Nepal experiences COVID 19, death, morbidity, mortality trends from COVID 19. Similarly, the paper did not discuss various published papers on similar topics in Nepal. Therefore requesting you to review more literature, rewrite the section. Give an overview of Nepal, why it is important to study this topic in Nepal. Also, think about whether the objective you have stated in Abstract and Introduction is same or different?

3. Methods: describe details about What is Mixed Method Cross Sectional Study? Basis for selecting 536 participants from seven provinces of Nepal? Sampling methods should be clear.

Line 83: How the list was obtained through a written communication or verbal consent or any other methods? In the list how many participants were there? Did you select an equal number of symptomatic or asymptomatic participants?

Line 93: What is a qualitative survey? Please describe in detail. Also provide information about the validity of the tool you used for/ approaches to validate the tool?

Line 96: Please cite (ERB reference number).

Line 101: what do you mean by similar setting. Describe in detail.

Also provide the details of 25 participants of qualitative study such as Police Y#, Health worker ##, Farmer ###.

I was also surprised to know physical, mental and social health was explored but what is the standard tool used for it. Provide all of your tools in both Nepali and English versions in Annex.

Also, it is important to discuss who performed the qualitative analysis. How it was done, how the highest degree of standards were met?

The section also did not talk about the variable used in the study.

4. Results: The whole section lacks interpretation and fluency in writing. Qualitative information is presented but lacks connection with quantitative information throughout the section. Different categories of people have been interviewed and it is important to present the finding- was it same to all the professions or the seriousness was different for a particular profession.

5. Discussion: It is interesting to know that your paper helps to develop specific intervention policies and improves mental, social and physical health. Please provide what are the overall findings of your study in the first paragraph of the discussion. I also read the whole section but it made it difficult to understand what is physical, mental and social health and how it was impacted by COVID 19 or how it was among the COVID 19 recovered in Nepal. The whole section should be rewritten with appropriate referencing. Make the consistency for referencing throughout the paper.

I suppose there are definitely other papers in the similar field in Nepal involving quarantined persons, hospitalized persons etc. Please review and compare their findings with your own.

Limitation and strength of your study should be discussed clearly. I found you have written something in the conclusion section however you are also advised to present your strength.

6. Results: The section should be rewritten. Please review and rework on your conclusion section. Please validate your conclusion with Abstract.

Additional comments:

Table 3. Please use consistency in table building and data presentation. Please use symbols such as N and n correctly.

Authors’ contribution is very important and advised to mention in the paper. How the analysis, review, write up and finalization was done? Please also maintain consistency while writing authors’ degrees too.

Also, advised to check the whole data presentation and table once.

Reviewer #2: Dear authors,

thank you very much for the opportunity to review the submitted manuscript. The manuscript includes a survey about the health status of patients in recovery after Covid-19 infection.

I am appreciating this work and have got some minor and serious concerns.

SERIOUS CONCERNS

Section abstract, method and findings: I am confused about the method and findings. You report a mixed method study including quantitative and qualitative surveys. Hence, I would expect one part with quantitative results, and a second part with qualitative findings. Both results and findings, seemed to be mixed and it is irritating that qualitative findings are reported in a quantitative manner. I suggest to separate between both approaches, and report the results/findings independently. In section conclusion, you may combine both approaches together.

Page 3, Line 58: I do not understand the definition of recovery. On one hand, the period of recovery is defined by the severity of illness. So I would assume, recovery takes so much time till the severity of illness is less (who defines this?). Next sentence: according to WHO, recovery takes up to six weeks to fully recover (what is not the same as reduced severity of illness, by the way). In line 35 you report that after 2-3 weeks of recovery (according to your definition: this should be week 8-9), patients experience symptoms – but patients should have been recovered already, so how can they experience symptoms? So, what is recovery: partly or full recovery from an illness, or just the end of symptoms, or alleviation of severity of illness? Who defines this, e.g. physicians, patients, health insurance companies? Basically, there is no clear definition of recovery and I strongly recommend to add a literature based definition, to improve the understanding of your research.

P4, L80: this is a strong argument, creating representative data for whole Nepal. Please provide in section methods a sufficient power calculation and report the results in 95% confidence intervals (or 90% or 99%, if you prefer). Provide information that your sample is representative for the whole population in Nepal (I assume that not every second person ≥18 years old is a police or military person, hence something might have gone wrong), or just delete this sentence in line 80.

Section methods: please use some headlines, e.g. setting, inclusion criteria, exclusion criteria, recruitment, information sources, analysis. Since you are performing a mixed method study, you should report this for both approaches.

MINOR CONCERNS

Page 2, Line 26 The sentence “following recovery …” communicates the same information as the sentence before, and can be deleted.

P2, L48 the words “friendly environment in quarantine, isolation, and hospital settings” have no relation to sections findings. Please report only conclusions, which are based on methods and findings/results. Hence, add some findings about quarantine, isolation, and hospital settings, or delete the conclusion.

P3, L60 This is an unusual term “affected by … the hospital environment”. What do you mean here? Are still in hospital? Or the rehabilitation s affected by the hospital environment, e.g. walls, rooms, furniture?

P4, L90: please, report more information about the questionnaire, e.g. categories, number of questions, types of answers and others.

P7, L163: I am confused about this information – where do the answers come from? Please report more in section methods, so that readers can understand the results.

P7, L174 Family perceptions? How can this be? It should read “how participants perceived their families”

Section results and discussion is wordy, but without major flaws.

Part limitations belongs before conclusions.

6. PLOS authors have the option to publish the peer review history of their article (what does this mean?). If published, this will include your full peer review and any attached files.

Reviewer #1: **Yes: **Kanchan

Reviewer #2: No

---

## [Author Response · Author response to Decision Letter 0]

4 Jul 2023

Reviewer: 1. 

Comment 1: Your way of referencing is varied throughout the paper. Please refer to PLOS one referencing guideline for details. Please make consistency for referencing throughout the paper.

Author response: Thank you so much for pointing out this important issue. We have revised and updated references according to the guideline of Plos one. 

Comment 2. Introduction section is completed within a paragraph which still can provide a lot of background information about Nepal. Though little information of Nepal is given. Still, it is important to discuss how Nepal experiences COVID 19, death, morbidity, mortality trends from COVID 19. Similarly, the paper did not discuss various published papers on similar topics in Nepal. Therefore, requesting you to review more literature, rewrite the section. Give an overview of Nepal, why it is important to study this topic in Nepal. Also, think about whether the objective you have stated in Abstract and Introduction is same or different?

Author response: We appreciate you identifying important concerns and providing valuable feedback. Agreeing on the suggestion we have revised and update the background section as per the feedback

 Regarding the objectives we have same objectives in whole study. 

Comment 3. Methods: describe details about What is Mixed Method Cross Sectional Study? Basis for selecting 536 participants from seven provinces of Nepal? Sampling methods should be clear.

Author response: We thank the reviewer for seeking further clarification on this. We have adopted descriptive cross-sectional study by using both qualitative and quantitative approaches. The total participants of the study were 552 and was selected purposively selected from list obtained from Epidemiology and disease control division. The detail is available in the manuscript. 

Comment 4 Line 83: How the list was obtained through a written communication or verbal consent or any other methods? In the list how many participants were there? Did you select an equal number of symptomatic or asymptomatic participants?

 Author response: Thank for raising the issues A written official letter was sent to Epidemiology and disease control division to obtain the list of the COVID -19 recovered patients. A total of 19,000 were available in the list. 

Comment 5: Line 93: What is a qualitative survey? Please describe in detail. Also provide information about the validity of the tool you used for/ approaches to validate the tool?

Author response: Thank you very much for the comment. We basically collected the data by using both qualitative and quantitative methods. In qualitative survey we used the open-ended questions to generate the in-depth answers.

For the quantitative questionnaire pretesting was conducted among the 10 recovered participants to obtain validation of the questionnaires. 

In qualitative part the tool was designed and revised several times by the expertise of related field to ensure its validity. 

Line 96: Please cite (ERB reference number).

Author response: ERB reference number is 569/2020

Line 101: what do you mean by similar setting. Describe in detail.

Author response: For the quantitative questionnaire we did the pretesting by collecting the data via phone among the 10 Covid recovered patients form the available list. 

Also provide the details of 25 participants of qualitative study such as Police Y#, Health worker ##, Farmer ###.

 Thank you for the comment it has been attached to the figure section. 

I was also surprised to know physical, mental and social health was explored but what is the standard tool used for it. Provide all of your tools in both Nepali and English versions in Annex

Author response: We didn’t used the standard tool. Questionnaire was designed by the expertise of the concerned field and with several consultation data was collected after pretesting. The tools are attached in Annex. 

Also, it is important to discuss who performed the qualitative analysis. How it was done, how the highest degree of standards was met?

Author response: In-depth Interview was conducted via telephonic interview. Audio recordings were than translated in English and were thoroughly checked for consistency. In the first round, researchers screened all the transcripts to create possible codes for the transcript. In the second phase, researchers thoroughly read the transcripts line by line to capture the real essence of data through open codes. Similar codes were grouped to reduce and narrow down the data. Themes were finalized with subsequent discussion with the research team to ensure that they are in accordance with the objectives of the study. Quotations, that were clear and express ideas that were relevant and interesting in terms of the purpose of the study were selected to cite as verbatim while presenting the findings. 

4. Results: The whole section lacks interpretation and fluency in writing. Qualitative information is presented but lacks connection with quantitative information throughout the section. Different categories of people have been interviewed and it is important to present the finding- was it same to all the professions or the seriousness was different for a particular profession. 

Author response: Thank you for bringing these concerns to our attention. We have edited the results section as per the feedback and triangulation have been done with in between the qualitative and qualitative data where ever possible. We have interviewed the different categories of the people similar findings have been found, only in army profession self-motivation was high so included that statement, but we have omitted this to reduce the confusion. 

5. Discussion: It is interesting to know that your paper helps to develop specific intervention policies and improves mental, social and physical health. Please provide what are the overall findings of your study in the first paragraph of the discussion. I also read the whole section but it made it difficult to understand what is physical, mental and social health and how it was impacted by COVID 19 or how it was among the COVID 19 recovered in Nepal. The whole section should be rewritten with appropriate referencing. Make the consistency for referencing throughout the paper.

I suppose there are definitely other papers in the similar field in Nepal involving quarantined persons, hospitalized persons etc. Please review and compare their findings with your own.

Limitation and strength of your study should be discussed clearly. I found you have written something in the conclusion section however you are also advised to present your strength.

Strength of the study:

Author response: Thank you for the comment. We have addressed all the mentioned comments in the manuscript. 

6. Results: The section should be rewritten. Please review and rework on your conclusion section. Please validate your conclusion with Abstract.

Author response: Thank you so much for comments, we have reviewed the result section and edited it as per the feedback. 

Additional comments:

Table 3. Please use consistency in table building and data presentation. Please use symbols such as N and n correctly. Also, advised to check the whole data presentation and table once.

Author response: Thank you for bringing this issue to our attention. we have maintained consistency in table building and data presentation. 

Authors’ contribution is very important and advised to mention in the paper. How the analysis, review, write up and finalization was done? Please also maintain consistency while writing authors’ degrees too.

Author response: Thank you for the comments. we have addressed the issue in the manuscript. 

Reviewer #2: Dear authors,

thank you very much for the opportunity to review the submitted manuscript. The manuscript includes a survey about the health status of patients in recovery after Covid-19 infection.

I am appreciating this work and have got some minor and serious concerns.

SERIOUS CONCERNS

Section abstract, method and findings: I am confused about the method and findings. You report a mixed method study including quantitative and qualitative surveys. Hence, I would expect one part with quantitative results, and a second part with qualitative findings. Both results and findings, seemed to be mixed and it is irritating that qualitative findings are reported in a quantitative manner. I suggest to separate between both approaches, and report the results/findings independently. In section conclusion, you may combine both approaches together.

Author response: Thank you for pointing out serious concern, these issues has also raised by the reviewer 1 and suggest us to do appropriate connection so we tried to separate the data in both abstract as well as in result section, hope you will be clear by the edited version, also we have triangulated the quantitative and qualitative findings so that readers could easily perceive them. 

Page 3, Line 58: I do not understand the definition of recovery. On one hand, the period of recovery is defined by the severity of illness. So, I would assume, recovery takes so much time till the severity of illness is less (who defines this?). Next sentence: according to WHO, recovery takes up to six weeks to fully recover (what is not the same as reduced severity of illness, by the way). In line 35 you report that after 2-3 weeks of recovery (according to your definition: this should be week 8-9), patients experience symptoms – but patients should have been recovered already, so how can they experience symptoms? So, what is recovery: partly or full recovery from an illness, or just the end of symptoms, or alleviation of severity of illness? Who defines this, e.g. physicians, patients, health insurance companies? Basically, there is no clear definition of recovery and I strongly recommend to add a literature-based definition, to improve the understanding of your research.

Author Response: Thank you for raisings serious issue , We have omit all the confusion part and as per the questionnaire we gave enrolled the recovered participants of past 4 weeks by being based on the CDC definition of Post recovery which has been added in our manuscript. 

P4, L80: this is a strong argument, creating representative data for whole Nepal. Please provide in section methods a sufficient power calculation and report the results in 95% confidence intervals (or 90% or 99%, if you prefer). Provide information that your sample is representative for the whole population in Nepali (I assume that not every second person ≥18 years old is a police or military person, hence something might have gone wrong), or just delete this sentence in line 80.

Author response: Thank you for the highlighting the issues. Since this study is descriptive in nature, as well as no way inferential analysis have been carried out, in this regards we assume power calculation and reporting the interval estimates may not be logical. Furthermore, this study was conducted within the time frame of five months (February to June 2020) in a pandemic situation on request of MoHP (Ministry of the health and population). During that period the data was collected via phone interview among those participants who were able to contact and willing to give the answer. 

Section methods: please use some headlines, e.g., setting, inclusion criteria, exclusion criteria, recruitment, information sources, analysis. Since you are performing a mixed method study, you should report this for both approaches.

 Author response: Thank you for your remark. In response to the insightful advice you offered, we have added a distinct title and divided the methodology part into its own section with the necessary heading as per the feedback. 

MINOR CONCERNS

Page 2, Line 26 The sentence “following recovery …” communicates the same information as the sentence before, and can be deleted.

Author response: Thank you so much for the feedback, it has been deleted as per suggestion.

P2, L48 the words “friendly environment in quarantine, isolation, and hospital settings” have no relation to sections findings. Please report only conclusions, which are based on methods and findings/results. Hence, add some findings about quarantine, isolation, and hospital settings, or delete the conclusion.

Author response: Thank you for the valuable comments, we have rewritten the whole introduction section. 

P3, L60 This is an unusual term “affected by … the hospital environment”. What do you mean here? Are still in hospital? Or the rehabilitations affected by the hospital environment, e.g. walls, rooms, furniture?

Author response: Thank you for the valuable comments, we have rewritten the whole introduction section. 

P4, L90: please, report more information about the questionnaire, e.g. categories, number of questions, types of answers and others.

Author response: Thank you very much for the comment. We have added more to it on the data collection tools and technique sections of the manuscript. 

P7, L163: I am confused about this information – where do the answers come from? Please report more in section methods, so that readers can understand the results.

P7, L174 Family perceptions? How can this be? It should read “how participants perceived their families”

Author response: Thank you for the pointing out these issues, we have corrected as per the suggestions.

---

## [Decision Letter · Decision Letter 1]

15 Aug 2023

Physical, Mental and Social status after COVID -19 Recovery in Nepal: A Mixed Method Study

PONE-D-22-22288R1

Dear Dr. Silwal,

We’re pleased to inform you that your manuscript has been judged scientifically suitable for publication and will be formally accepted for publication once it meets all outstanding technical requirements.

Kind regards,

Rosemary Frey

Academic Editor

PLOS ONE

Additional Editor Comments (optional):

Reviewers' comments:

Reviewer's Responses to Questions

**Comments to the Author**

1. If the authors have adequately addressed your comments raised in a previous round of review and you feel that this manuscript is now acceptable for publication, you may indicate that here to bypass the “Comments to the Author” section, enter your conflict of interest statement in the “Confidential to Editor” section, and submit your "Accept" recommendation.

Reviewer #1: All comments have been addressed

Reviewer #2: All comments have been addressed

2. Is the manuscript technically sound, and do the data support the conclusions?

Reviewer #1: Yes

Reviewer #2: Yes

3. Has the statistical analysis been performed appropriately and rigorously? 

Reviewer #1: I Don't Know

Reviewer #2: N/A

4. Have the authors made all data underlying the findings in their manuscript fully available?

Reviewer #1: Yes

Reviewer #2: Yes

5. Is the manuscript presented in an intelligible fashion and written in standard English?

Reviewer #1: Yes

Reviewer #2: Yes

6. Review Comments to the Author

Reviewer #1: Dear Authors,

Thank you for addressing all the relevant concern raised during the peer review process. I believed that the findings will definitely add value in the field of public health globally with evidence from Nepal.

Reviewer #2: Dear authors

Thank you very much for the revision of the manuscript. You were able to address all of my concerns, or argued reasonable. I have no further concerns. Well done.

7. PLOS authors have the option to publish the peer review history of their article (what does this mean?). If published, this will include your full peer review and any attached files.

Reviewer #1: **Yes: **Kanchan Thapa

Reviewer #2: No

---

## [Editor Report · Acceptance letter]

24 Aug 2023

PONE-D-22-22288R1 

Physical, Mental and Social status after COVID -19 Recovery in Nepal: A Mixed Method Study 

Dear Dr. Silwal:

I'm pleased to inform you that your manuscript has been deemed suitable for publication in PLOS ONE. Congratulations! Your manuscript is now with our production department. 

Kind regards, 

on behalf of

Dr. Rosemary Frey 

Academic Editor

PLOS ONE